# Fluorescent Anti-CEA Nanobody for Rapid Tumor-Targeting and Imaging in Mouse Models of Pancreatic Cancer

**DOI:** 10.3390/biom12050711

**Published:** 2022-05-16

**Authors:** Thinzar M. Lwin, Michael A. Turner, Hiroto Nishino, Siamak Amirfakhri, Sophie Hernot, Robert M. Hoffman, Michael Bouvet

**Affiliations:** 1Department of Surgery, University of California San Diego, San Diego, CA 92093, USA; helmi.lwin@gmail.com (T.M.L.); maturner@health.ucsd.edu (M.A.T.); hnishino@kuhp.kyoto-u.ac.jp (H.N.); siamirfakhri@health.ucsd.edu (S.A.); meishale@gmail.com (R.M.H.); 2Department of Surgical Oncology, Dana Farber Cancer Institute, Boston, MA 02215, USA; 3Department of Surgery, VA San Diego Healthcare System, San Diego, CA 92161, USA; 4Laboratory for In vivo Cellular and Molecular Imaging, ICMI-BEFY/MIMA, Vrije Universiteit Brussel, B-1090 Brussels, Belgium; sophie.hernot@vub.be; 5AntiCancer, Inc., San Diego, CA 92111, USA

**Keywords:** nanobodies, pancreatic cancer, patient derived orthotopic xenograft, CEA, fluorescence-guided-surgery, fluorescence, near-infrared, IRDye800CW, tumor labeling

## Abstract

Tumor-specific targeting with fluorescent probes can enhance contrast for identification of cancer during surgical resection and visualize otherwise invisible tumor margins. Nanobodies are the smallest naturally-occurring antigen-binding molecules with rapid pharmacokinetics. The present work demonstrates the efficacy of a fluorescent anti-CEA nanobody conjugated to an IR800 dye to target and label patient derived pancreatic cancer xenografts. After intravenous administration, the probe rapidly localized to the pancreatic cancer tumors within an hour and had a tumor-to-background ratio of 2.0 by 3 h. The fluorescence signal was durable over a prolonged period of time. With the rapid kinetics afforded by fluorescent nanobodies, both targeting and imaging can be performed on the same day as surgery.

## 1. Introduction

The goal of oncologic surgery is to obtain complete resection of all tumors. Despite the best attempts at pre-operative localization, surgeons still rely on subjective visual inspection, palpation, intra-operative frozen sections, and clinical judgment to determine completeness of resection and achieve this goal [1]. Visual and tactile cues may miss small metastases, especially sub-centimeter lesions not detectable by pre-operative imaging modalities. Frozen sections are subject to sampling error as there is a large surface area to sample from. This can be further exacerbated by the increased trend in the use of minimally-invasive resection for pancreatic cancer and the increasing use of neoadjuvant chemoradiation [2,3]. With a minimally-invasive approach, tactile feedback is lost, and assessment relies primarily on visual cues [4,5]. With neoadjuvant therapies, previously treated areas of scarring and fibrosis are indistinguishable from cancer. The naked eye with white light is insufficient to detect a contrast between cancer and normal tissue, especially when tumors can often have a similar color and texture as adjacent tissue. An objective labeling method with a contrast agent can better enhance intraoperative clinical decision making. 

Tumor-specific labeling with probes bearing fluorophores, detectable by near-infrared (NIR) intraoperative imaging devices could be a potential solution [6]. Contrast-enhanced visualization and molecular labeling can assist in visualization of pancreatic cancer and, if present, any intra-abdominal metastases that would preclude the patient from an invasive and likely ineffective surgical procedure.

The human carcinoembryonic antigen (CEA) is a well-studied tumor biomarkers in pancreatic cancer and is overexpressed in 83–98% of pancreatic cancers [7]. CEA is a cell-surface glycoprotein that is normally expressed during embryogenesis, with little to no expression in normal tissue, and is used as a serum tumor marker in patients with pancreatic cancer [8]. Anti-CEA antibodies linked to NIR fluorophores have been used to label pancreatic cancer in pre-clinical and early clinical studies [9,10,11]. However, due to the large size of intact antibodies (~150 kDa), there is decreased target penetration efficacy and a delay in time to peak signal. This becomes an important issue in the timing of surgery as there is a significant delay between administration of an antibody-fluorophore conjugate and imaging, leading to the need for patients to present for a separate clinical visit for probe administration. 

Fluorescent probes derived from nanobodies, the smallest naturally occurring antigen binding molecules, can deliver a fluorescence signal to the tumor within hours [12]. Nanobodies^®^ (Ablynx/Sanofi) are single-domain variable fragments of 15 kDa [13]. At one-tenth the size of traditional antibodies, these molecules retain the affinity of antibodies, but can more rapidly penetrate and bind antigens [14,15]. Furthermore, nanobodies are more pH stable, heat-stable, and resistant to proteolytic degradation, making them a promising platform for tumor-specific fluorophore delivery [16]. In a previous study from our laboratory, anti-CEA nanobodies conjugated to the near-infrared dye IRDye800CW were used to target and label pancreatic cancer cell-line derived orthotopic xenograft mouse models [17]. It was found that the fluorescent anti-CEA nanobody could target and label the tumor very rapidly. In the current work, we studied the tumor-persistence of the fluorescent anti-CEA nanobody over a prolonged time period in a patient-derived orthotopic xenograft (PDOX) mouse model of pancreatic cancer. 

## 2. Materials/Methods

Nanobody synthesis and conjugation: Previously-generated anti-CEA (NbCEA5) or control nanobodies (R3B23) were produced as described previously [18,19]. The control nanobody targets the M-protein expressed by 5TMM in multiple myeloma mice [20]. This nanobody does not recognize any target in healthy mice and is, therefore, used as a control nanobody in our experiments. Nanobody constructs with a carboxy-terminal cysteine tag were cloned into a pHEN6c plasmid and expressed in *E. coli*. Nanobody products were purified from peri-plasmic extracts using immobilized metal affinity chromatography followed by subsequent size-exclusion chromatography. The purified nanobodies (1 mg/mL) were reduced using the mild reductant 2-mercaptoethylamine (2-MEA) Hydrochloride (Acros Organics, Geel, Belgium) at 180-fold molar excess and 5 mM EDTA over 90 min at 37 °C. Following removal of 2-MEA using a PD-10 desalting column (GE Healthcare, Chicago, IL, USA) equilibrated with PBS, the reduced nanobodies were incubated for 2 h at 37 °C with a 5-fold molar excess of maleimide-derivatized IRDye800CW (LI-COR Biosciences, Lincoln, NE, USA) and 5 mM EDTA. The fluorescent nanobodies termed aCEA-nb-800 and aCtrl-nb-800, were subsequently purified by size-exclusion chromatography on a Superdex 75 10/300 GL column (GE Healthcare, Chicago, Il, USA) with PBS as the elution buffer (0.5 mL/min). Detection at 280 nm and 774 nm was used for analysis of protein and dye absorbance, respectively. The concentration of the labeled nanobodies and the degree of fluorophore conjugation were determined by measuring absorbance of the protein and the dye with a UV−Vis spectrophotometer (Nanodrop2000, Thermo-Fisher, Waltham, MA, USA), taken a 3%-correction factor for absorbance of the dye at 280 nm into account. The degree of labeling was 0.9.

Animal Care: Immunocompromised nude nu/nu mice were maintained in a barrier facility on high-efficiency particulate air (HEPA)-filtered racks (VA Medical Center, San Diego, CA, USA). Mice were maintained on an autoclaved low-fluorophore laboratory rodent diet (Envigo, Indianapolis, IN, USA) and kept on a 12 h light/12 h dark cycle. Four-week-old male or female nude mice were used for tumor implantation. All surgical procedures and intravital imaging were performed with the animals anesthetized by intramuscular injection of an anesthetic cocktail composed of ketamine 100 mg/kg and buprenorphine 0.1 mg/kg. All animal studies were conducted in accordance with the principles and procedures outlined in the NIH Guide for the Care and Use of Animals under PHS Assurance Number D16-00394 (A3659-01) and A3033-1.

Patient derived orthotopic mouse models: Fresh pancreatic tumors were obtained during surgery from an IRB approved protocol UCSD IRB # 140046 (Approval date 28 September 2022). Pancreatic cancer tissue fragments were implanted subcutaneously into nude mice to establish subcutaneous patient-derived xenografts (*n* = 10). In an additional group of nude mice, tumor fragments were implanted into the pancreatic tail of nude mice (*n* = 4) using the technique of surgical orthotopic implantation (SOI) to establish PDOX models [21]. 

Fluorescence imaging of subcutaneous tumors over time: After the tumors were allowed to engraft for 4–6 weeks, 2 nmol of aCEA-nb-800 were injected intravenously (*n* = 10). The dose was selected as optimal based on previous in-vivo studies and is concordant with other studies using nanobody-fluorophore conjugates [15,22,23]. Mice were serially imaged over time using the Pearl Trilogy small animal imager (LI-COR Biosciences, Lincoln, NE, USA). Mice were kept continually monitored, and kept warm and under complete anesthesia for the duration of the procedure. Mice were monitored for muscular tone and response to stimulation, and rate and depth of respiration and anesthesia was re-dosed appropriately. All imaging experiments were performed in a light tight box with the same imaging acquisition settings (i.e., excitation, emission channels, and exposure and/or gain of detectors). Maximal fluorescence intensity (MFI) is measured from designated regions of interest (ROI) in arbitrary units. The tumor-to-background ratio (TBR) was calculated by dividing the MFI recorded from the tumor divided by the MFI from the surrounding background pancreatic tissue. Approaches to measurement of ROI are indicated in Appendix A.

Fluorescence imaging of orthotopic tumors over time: PDOX models were established and were allowed to engraft for 4–6 weeks (*n* = 4). The PDOX mouse models were labeled with 2 nmol of either aCEA-nb-800 or aCtrl-nb-800. Images were taken after necropsy using the LICOR-Odyssey (LI-COR Biosciences, Lincoln, NE, USA). Images were obtained under white light and fluorescence (800 nm) excitation sources. Fluorescence intensity was mapped using the LICOR Image studio software to generate a color heat-map image.

Data are represented as mean ± standard deviation. The maximal fluorescence intensity (MFI) and the TBR were plotted over time, error bars represent standard error of the mean (SEM = standard deviation/Sqrt (counts)). The rate of decay of maximal fluorescence intensity was compared over time for tumor and background. The initial signal at time 1 h was compared to all other subsequent time points as a percentage of original and tabulated. 

## 3. Results

A fluorescence signal was detectable at patient-derived subcutaneous tumors within an hour of intravenous injection. MFI from the tumor, the kidneys, and surrounding background tissue was monitored over 12 h (Figure 1). Serial images of a representative subcutaneous tumor are shown in Figure 1a. The fluorescence signal was between 0.9–1.8 a.u. over the 12 h time period that the mice were monitored. The mean MFI from the tumor peaked at 2 h (1.8 a.u. ± SEM 0.18) and decreased over subsequent time points. The mean MFI of the background peaked at 1 h (0.95 a.u. ± SEM 0.07) and decreased over subsequent time points. The mean MFI from the kidney peaked at 2 h (4.6 a.u. ± SEM 0.29) and gradually decreased over subsequent time points. The initial tumor to background ratios (TBR) were 1.9 at 1 h, 2.0 at 2 h, and 2.2 at 3 h. The peak TBR was at 6 h (2.9 ± SEM 0.49). The background fluorescence signal was maximal by the first hour and the signal decreased over the time period that the mice were monitored. The tumor fluorescence signal was maximal by the second hour and its signal decreased over the time period that the mice were monitored. There was a more rapid decrease in background fluorescence compared to tumor fluorescence (Table 1). More than 50% of the initial fluorescence signal was retained at the tumor after 8 h of monitoring (Table 1). 

In a pancreatic cancer PDOX mouse model, aCEA-nb-800 labeled small sub-centimeter tumors within 3 h of probe administration (Figure 2, top panel). Mean MFI at the tumor using aCEA-nb-800 was 1.9 a.u. with a background MFI of 0.86 and a TBR of 2.2. Mean MFI from the tumor using aCtrl-nb-800 was 0.70 a.u. with a background MFI of 0.52 and a TBR of 1.3 (Figure 2, bottom panel). 

The pancreatic PDOX tumor was bivalved and a tissue section was further imaged ex-vivo using the LICOR-Odyssey (Figure 3). Images of the tumors labeled with aCEA-nb-800 and aCtrl-nb-800 were obtained under white light and fluorescence, and a pseudo-color intensity map was constructed. The fluorescence signal was localized at the tumor and did not show a signal from the surrounding pancreatic parenchyma. Under magnification, the tumor showed a TBR of 11.97 at the tumor using aCEA-nb-800 and a TBR of 2.97 using the aCtrl-nb-800 (Figure 3).

## 4. Discussion

The aCEA-nb-800 rapidly and successfully labeled patient derived xenografts of pancreatic cancer. In the subcutaneous model, there was a fluorescence signal at the tumor within 1–2 h of injection. Nanobodies are small molecules with a single valent binding site. In using them as fluorescence labeling probes, they are appealing as agents for obtaining a rapid tumor-specific signal. Studies have investigated the dose needed for sensitive and high contrast imaging with fluorescent nanobodies [15,24]. In previous work directly using orthotopic xenograft mouse models of colorectal cancer, optimal doses of this probe were evaluated and 2 and 3 nmol doses provided similar fluorescence intensity [22]. Therefore, the 2 nmol dose was selected for this study. In measuring the fluorescence signal, we opt to use maximal fluorescence intensity as it is independent from an ROI, less observer dependent and more reproducible compared to a mean fluorescence intensity [25].

The extended kinetics of this study demonstrate that there was clearance of both the tumor and the background fluorescence signal over time. However, the tumor-specific signal decreased slower over time compared to the background signal (Table 1). This drove the increase in TBR, and peak TBR was seen at hour 6. The tumors retained more than 94–95% of the initial fluorescence intensity in the first 5 h compared to the background which had 65% of initial fluorescence intensity at hour 5. This provides a respectable timeframe for same-day tumor labeling and imaging, even for longer cases. The optimal balance of contrast versus fluorescence intensity remains to be determined in future studies, as waiting for the optimal contrast may lead to a slightly decreased fluorescence intensity. 

The TBR of the aCEA_nb-800 probe ranged from 1.9–2.2 within the first 3 h of imaging, which is slightly lower than a mean TBR of 2.66 seen in previous work performed using BxPC-3 cell-lines with a high expression of CEA [17]. Studies on tumor-specific fluorescent probes have generally been performed using cell lines over-or under-expressing the target antigen as a proof of principle [26]. The tumor microenvironment is a heterogenous population and the ability of fluorescent antibodies to deliver an intense fluorescent signal could be affected by this. PDOX models mimic the heterogeneity of the tumor population and better represent the physiology and natural biology of the disease, making them a clinically relevant model for optical fluorescence imaging [27,28]. Based on these differences in pre-clinical and clinical studies, best practices for reporting on emerging optical imaging agents recommend a TBR > 3 for pre-clinical studies and a TBR > 1.5 for clinical studies [25]. In the present study using a patient derived tumor that expresses CEA, even with tissue heterogeneity, TBR values were greater than 1.5 throughout the time period evaluated. In clinical trials of SGM101, an anti-CEA antibody-based probe which has advanced the furthest clinically, a mean TBR value of 1.6 in patients with pancreatic cancers [29]. This value was clinically relevant in these studies as this degree of contrast enhancement permitted surgeons to identify tumors that they would otherwise have been missed. 

The TBR range produced by a nanobody is not as high as that obtained by full length antibodies which can carry more fluorophores per antibody. The nanobody is linked to the fluorophore using site-specific conjugation which optimizes binding and limits steric hinderance [18,30]. This can limit the maximal total number of fluorophores that the probe can carry using this conjugation technique. A higher degree of labeling of fluorophores per nanobody is a potential strategy to increase the fluorescence intensity of nanobody tracers. Although this is commonly applied for antibodies, this is less evident for smaller compounds such as nanobodies because of their small size (15 kDa). Conjugating 2 or more dyes could result in two fluorophores in close proximity and could lead to quenching of the fluorescent signal or altered biodistribution of the probe [31,32]. Alternative positioning of fluorophores, fluorophores even in the NIR-II range, or even activatable fluorescent probes that limit non-specific background signal could potentially be explored in the future to overcome this limitation [33,34,35,36]. 

This probe was still able to deliver a signal detectable by clinical NIR fluorescence imaging devices, such as the da Vinci Firefly and the Stryker AIM [17]. These devices have a lower sensitivity as they are designed to detect milligrams of non-targeted fluorescent dyes like indocyanine green. A number of operating rooms already have these fluorescence imaging systems and the use of spectral overlap between indocyanine green and IRDye800CW avoids the need to purchase additional costly imaging devices for tumor-specific fluorescence imaging [6]. Further refinement to adjust sensitivities can optimize signal detection. Even with these limitations, there was a detectable tumor-specific signal at the PDOX pancreatic cancer as early as 1 h, persisting for up to 12 h. The present study demonstrates proof-of-principle that a patient could be administered a fluorescent nanobody tracer on the same day as surgery which does not seem feasible at present with current antibody-based probes. 

## 5. Conclusions

Fluorescent aCEA-nb-800 specifically labeled PDOX pancreatic tumors. The use of a PDOX model allowed for stromal heterogeneity, which better mimics a clinically-relevant tumor microenvironment. In this setting, the fluorescent nanobody was able to deliver a tumor-specific signal with durability over hours. The kinetics of nanobodies allow for same day administration and imaging. Anti-CEA-nb-800 is a promising and practical molecule for FGS of pancreatic cancer.

## Figures and Tables

**Figure 1 biomolecules-12-00711-f001:**
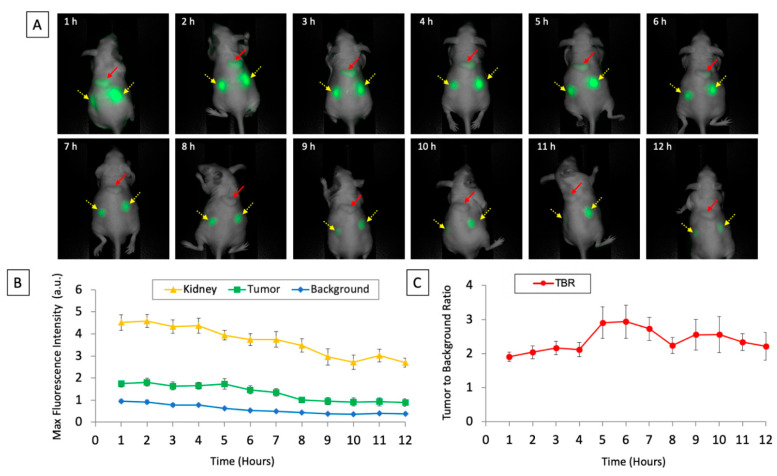
(**A**) Images of subcutaneously implanted patient-derived xenografts of pancreatic cancer after intravenous injection of aCEA-nb800 over the 12 timepoints measured using the LICOR Pearl small animal imager. The red arrow indicates the tumor and the yellow arrows with dashed lines indicate the kidneys. (**B**) Maximal fluorescence intensity of tumor, kidney, and background plotted over time. (**C**) Calculated tumor-to-background ratio from the tumor is plotted over time. Error bars indicate standard error.

**Figure 2 biomolecules-12-00711-f002:**
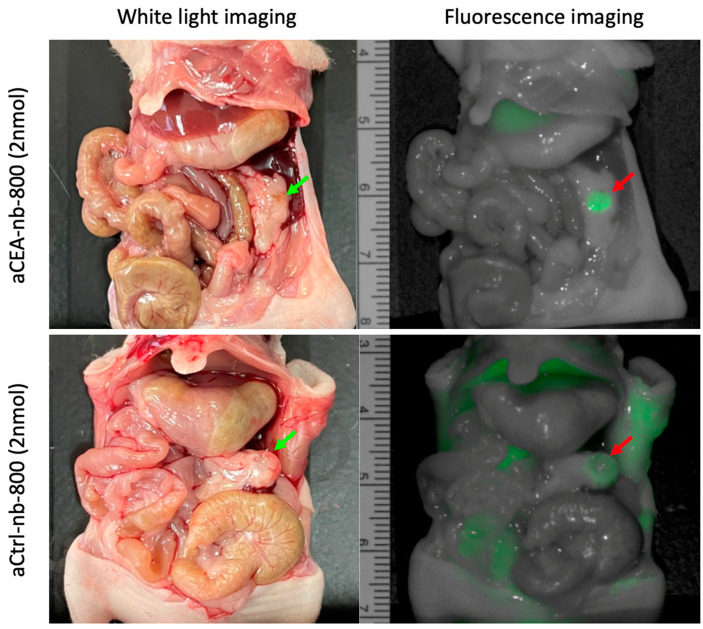
Pancreatic cancer patient-derived orthotopic xenograft (PDOX) mouse models labeled with either aCEA-nb-800 (**top** panels) or aCtrl-nb-800 (**bottom** panels). Mean MFI from the tumor using aCEA-nb-800 was 1.9 a.u. with a background MFI of 0.86 and a TBR of 2.2. Mean MFI from the tumor using aCtrl-nb-800 was 0.70 a.u. with a background MFI of 0.52 and a TBR of 1.3.

**Figure 3 biomolecules-12-00711-f003:**
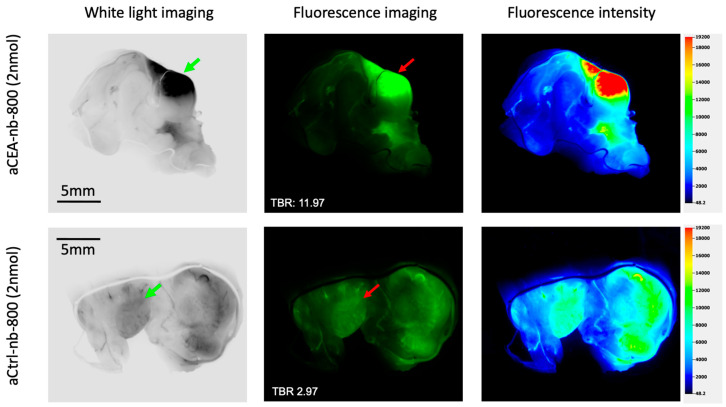
Pancreatic cancer orthotopic patient-derived xenograft labeled with either aCEA-nb-800 (**top** panels) or aCtrl-nb-800 (**bottom** panels), bivalved and imaged ex-vivo using the LICOR-Odyssey. Images were acquired under white light and fluorescence (800 nm) excitation sources. The fluorescence signal was localized at the tumor and did not show a signal at the surrounding pancreatic parenchyma. Under magnification, the tumor showed a TBR of 11.97 at the tumor using aCEA-nb-800 and a TBR of 2.97 using aCtrl-nb-800. Fluorescence intensity was mapped using the LICOR Image studio software to generate a color heat-ma.

**Table 1 biomolecules-12-00711-t001:** The rate of decrease of maximal fluorescence intensity was compared over time for tumor and background. The initial signal at 1 h was compared to all other subsequent time points as a percentage of original and tabulated below.

Time	% of Initial Signal (Tumor)	% of Initial Signal (Background)
2 h	103.96%	96.41%
3 h	94.22%	81.28%
4 h	95.43%	82.19%
5 h	99.68%	65.44%
6 h	84.28%	56.14%
7 h	77.76%	50.77%
8 h	57.37%	45.73%
9 h	54.98%	39.27%
10 h	52.88%	37.01%
11 h	53.24%	40.93%
12 h	51.13%	40.22%

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
