# Peer review of "Fluorescent Anti-CEA Nanobody for Rapid Tumor-Targeting and Imaging in Mouse Models of Pancreatic Cancer"

_biomolecules, 2022, doi:10.3390/biom12050711_

Round 1

Reviewer 1 Report

The authors describe that the efficacy of a fluorescent anti-CEA nanobody conjugated to an IR800 dye to target and label patient derived pancreatic cancer xenografts. The manuscript was written very well. The study was well-organized and carefully performed. I do not find any criticism for the results and discussion.

For improving the manuscript, the topics about activatable probes or activity-based probes that could further enhanced the tumor-to-background ratio should be added to the discussion (or introduction) in order to explain more about the fluorescence based bio-imaging field of works.

No other issue exists. A great work.

Author Response

We appreciate the feedback from the reviewer on this paper and appreciate the opportunity to strengthen specific aspects of this manuscript. We have added mention of “activatable fluorescent probes that limit non-specific background signal” which could potentially be explored in the future as part of our discussion. For the statement we have revised, we have added the reference: Mochida, Bioorganic & Medical Chemistry 2018, https://doi.org/10.1016/j.bmc.2017.12.002.

Reviewer 2 Report

The work of Lwin and co-authors with the title: “Fluorescent anti-CEA nanobody for rapid tumour-targeting and imaging in a patient-derived orthotopic xenograft mouse model of pancreatic cancer” focuses on a straightforward and not particularly novel question. Although the objective of the work is simple allowing straightforward results, I think the manuscript has several flaws and weak points that need to be addressed before acceptance for publication. Bellow I addressed the most relevant issues.

Lines 74: I could not find the target of the control Nanobody “R3B23”. I will suggest expanding the whole method section with more details. Experiments described in a scientific paper should be reproducible. The first step toward this is to describe experimental procedures in detail and avoid citing previous work, where, often, there is also not possible to find the needed information.  

Lines ~78-80: About the labelling of nanobody. Since this is very relevant for the work,  I would consider a slightly more detailed procedure necessary, how was the nb reduced (time, chemicals use and concentration, temperature, buffers, etc.). In the same direction, a better characterization of the tool used seems appropriate, at least a measurement of the degree of labelling seems necessary. A poor degree of labelling will result in the need to inject more nanobodies into the bloodstream (also more endotoxins from bacteria), while a reasonable degree of labelling will require less and make a more sensitive assay. Why add one fluorophore only? Wouldn´t two fluorophores make the detection more sensitive?

Line 99: Why injected 2 nmol? This seems randomly chosen. This will result in ~1 µM of VHH, assuming a mouse has ~2 ml of blood, which is OK (depending on the Kd of the Nb). I am wondering why this number? Has it to do with the Nb affinity, or toxicity?

Line 111: “A strong signal of 0.7 to 1.4 a.u” without a reference. This seems to be a relatively weak signal, especially if this is the raw signal, not background subtracted (as it seems due to the Y axes in Figures 1a and 1b). “specific signal from the tumour is composed of background signal (from tissue) + specific signal.  

Line 113:  I understood Authors sampled every hour. What is the max signal at 0.5h? I mention this not to re-do the experiments but to rephrase your finding more carefully. Max signal was at 1h, because the authors did not look earlier?

Line 117-118: Now it seems that the authors did sample more often than every hour. Why is this not plotted? Or how suddenly come values at 15 minutes?

Figure 1:

  • Why is the background signal “0” at time 0? Background signal should be present always if time 0 means signal before injection of the fluorescence Nb. Strictly speaking fluorescence signal of all three categories, kidney, tumour and background, should be higher than “0” before nanobody injection.
  • It will be necessary that all experiments were performed with the same imaging acquisition settings, e.g., excitation energy and signal collection (exposure and/or gain of detectors)
  • Why does the background signal go down in time? Your TBR shows no change in time of the ratio fluorescence to the background. Could it be that this is bleaching, or drying? Would it not be more sensitive to subtract the background signal at each time point? Resulting in only a fluorescence component? This will show that the “real” fluorescent signal remains very constant over time. Image examples from first timepoint, mid and last will be useful/transparent to show.
  • An essential missing control is to monitor the same parameters but using a Control-nb-800. I expect the Kidney to display something similar to the aCea-nb-800, while the tumour should be lower and background comparable. Only then do you show a specific increase of signal at the tumour using the aCea-nb-800.
  • It is a sound scientific practice to add in figure legends the type of error displayed (in this case is Stdv, but the reader will also like to know the number of independent experiments to better judge the distribution of the variance.
  • Figure 1b how are these error bars calculated? There are specifics rules on how to deal when doing arithmetic with errors. Again, this should be explained/described. Looking at this curve, little can be concluded as errors are so large that there is basically no difference between any time point, thus no conclusions can be made from the graph as presented now.

Line 136-138: Are those values of a single mouse/measurement? If these are the values of the examples shown in Figure 2, could you show the reader what regions were selected for the analysis as “tumour” and “background” signals? A small dotted line showing the ROI will be helpful.

I understand that it might not be easy to use patient materials to generate an orthotopic xenograph, but a single experiment is still scientifically problematic, this could be chance, luck or artifactual.  If this is the case, I will suggest toning down the title and conclusions, since this has been shown only once, and even here with some questions.

Figure 3:

  • White light imaging? This looks more like a grey level of the fluorescence image. Please double-check, and if this is the same image iin another color scheme is not informative.
  • How do readers know that the arrow point to the tumour? Because the fluorescence signal is there? This seems to be a tautological explanation. In the images, there are other regions also rather bright, not designated as tumours, what are the criteria? Immunofluorescence or other methods is needed here to demonstrate that the fluorescence obtained by the nanobody coincides with the tumour using another biomarker.
  • Images need scale bars.
  • The only difference between “Fluorescence imaging” and “Fluorescence intensity” panels is the lookup table or LUT, but both are images and display fluorescence intensities. The “royal” lookup table only makes it easier to “see” the differences in intensity when compared to the “green” LUT. Good practice will be to display the LUT range beside the image to understand the colour-coding.

Line 170-171: Please rephrase this statement. The control nanobody did show a “specific” fluorescence signal at the tumour, it was just a lower signal (Figure 3 even has a dedicated arrow). Unfortunately, this (control nanobody) was not tested in the initial experiments performed for Figure 1, which would have helped here in the discussion.

Line115: Once again, reference to a 15-minute signal that was never shown within the manuscript.

Author Response

The work of Lwin and co-authors with the title: “Fluorescent anti-CEA nanobody for rapid tumour-targeting and imaging in a patient-derived orthotopic xenograft mouse model of pancreatic cancer” focuses on a straightforward and not particularly novel question. Although the objective of the work is simple allowing straightforward results, I think the manuscript has several flaws and weak points that need to be addressed before acceptance for publication. Bellow I addressed the most relevant issues.

 We appreciate the feedback from the reviewer on this paper and appreciate the opportunity to strengthen specific aspects of this manuscript. We have revised the manuscript and the figures extensively. We have addressed the reviewer concerns point by point below. We believe that the revised manuscript is suitable for publication.

Lines 74: I could not find the target of the control Nanobody “R3B23”. I will suggest expanding the whole method section with more details. Experiments described in a scientific paper should be reproducible. The first step toward this is to describe experimental procedures in detail and avoid citing previous work, where, often, there is also not possible to find the needed information.  

  • The control nanobody is a non-targeting nanobody that is specific for the M-protein expressed by 5TMM multiple myeloma mice (Lemaire, Leukemia, 2014, doi:10.1038/leu.2013.292 This information has been added to the revised manuscript under the materials/methods section: “The control nanobody is a non-targeting nanobody that targets the M-protein expressed by 5TMM multiple myeloma mice. This nanobody does not recognize any target in healthy mice and is therefore used as control nanobody in most of our experiments.”

Lines ~78-80: About the labelling of nanobody. Since this is very relevant for the work,  I would consider a slightly more detailed procedure necessary, how was the nb reduced (time, chemicals use and concentration, temperature, buffers, etc.). In the same direction, a better characterization of the tool used seems appropriate, at least a measurement of the degree of labelling seems necessary. A poor degree of labelling will result in the need to inject more nanobodies into the bloodstream (also more endotoxins from bacteria), while a reasonable degree of labelling will require less and make a more sensitive assay. Why add one fluorophore only? Wouldn´t two fluorophores make the detection more sensitive?

  • Further details on labeling of the nanobody have been added to the manuscript. “The purified nanobodies (1 mg/mL) were reduced using the mild reductant 2-mercaptoethylamine (2-MEA) Hydrochloride (Acros Organics) at 180-fold molar excess and 5 mM EDTA over 90 min at 37 °C. Following removal of 2-MEA using a PD-10 desalting column (GE Healthcare) equilibrated with PBS, the reduced Nanobodies were incubated for 2 h at 37 °C with a 5-fold molar excess of maleimide-derivatized IRDye800CW (LI-COR Biosciences) and 5 mM EDTA. The fluorescent Nanobodies were subsequently purified by size-exclusion chromatography on a Superdex 75 10/300 GL column (GE Healthcare) with PBS as elution buffer (0.5 mL/min). Detection at 280 and 774 nm was used for analysis of protein and dye absorbance, respectively. The concentration of the labeled Nanobodies and the degree of fluorophore conjugation were determined by measuring absorbance of the protein and the dye with a UV−Vis spectrophotometer (Nanodrop2000), taken a 3%-correction factor for absorbance of the dye at 280 nm into account. The degree of labeling was 0.9.”
  • We agree with the reviewer that a higher degree of labeling is an interesting strategy to increase the fluorescence intensity of tracers. Although this is commonly applied for antibodies, this is less evident for smaller proteins such as nanobodies. Because of their small size (15 kDa), conjugating 2 or more dyes could result in two fluorophores in close proximity and could lead to quenching of the fluorescent signal. Moreover, it is well known that the conjugation of fluorescent dyes as well as the degree of labeling can significantly impact the bio-distribution of the probe (Cilliers, Mol Pharm, 2017, https://doi.org/10.1021/acs.molpharmaceut.6b01091 and Debie, Mol Parm 2017, https://doi.org/10.1021/acs.molpharmaceut.6b01053). This discussion and references have been added to the revised manuscript under discussion.

Line 99: Why injected 2 nmol? This seems randomly chosen. This will result in ~1 µM of VHH, assuming a mouse has ~2 ml of blood, which is OK (depending on the Kd of the Nb). I am wondering why this number? Has it to do with the Nb affinity, or toxicity?

  • The anti-CEA nanobody has an affinity of 0.78 nM, which is slightly better than the affinity of most nanobodies used in vivo (typically low nanomolar range). Studies have investigated the dose needed for sensitive and high contrast imaging with fluorescent nanobodies (Debie, Antibodies, 2019, doi: 10.3390/antib8010012, Banas, Contrast Media Mol Imaging 2015, DOI: 10.1002/cmmi.1637). In these studies doses between 1-10 nmol lead to fluorescent signals that can be adequately detected by in vivo fluorescence imaging devices (including clinical systems for intraoperative imaging). Further increase of the dose will cause saturation of the signals.
  • In previous work using orthotopic xenograft mouse models of colorectal cancer, 1, 2, and 2 nmol doses were evaluated and the 2 and 3 nmol doses provided similar fluorescence intensity (Lwin, JSO, 2021, DOI: 10.1002/jso.26623). This information and references have been added to the revised manuscript under discussion.

Line 111: “A strong signal of 0.7 to 1.4 a.u” without a reference. This seems to be a relatively weak signal, especially if this is the raw signal, not background subtracted (as it seems due to the Y axes in Figures 1a and 1b). “specific signal from the tumour is composed of background signal (from tissue) + specific signal.  

  • We agree with the reviewers and have revised the sentence to read as follows: “The fluorescence signal was between 0.7-1.4 a.u. over the 12 hour time period that the mice were monitored.”
  • We agree that the specific signal from the tumor comprises both the background and the tumor specific signal. However, in evaluating fluorescent probes for tumor-labeling and fluorescence-guided surgery, it is conventional to report the tumor to background ratio as it is the contrast against surrounding tissue that provides the most clinically meaningful information.(Recommendations for reporting on emerging optical imaging agents to promote clinical approval (Tummers, Theranostics, 2018, doi:10.7150/thno.27384). This information and references have been added to the revised manuscript under methods and discussion.
  • We believe that the signals are clinically meaningful as they were detectable using clinical available fluorescence imaging devices (Lwin, Surgery, 2020, DOI: 10.1016/j.surg.2020.02.020). This information and reference have been added to the revised manuscript under discussion.
  • Clinical trials of SGM101, an anti-CEA antibody based fluorescent probe which has advanced the furthest clinically, showed an average TBR of 1.6 in patients. This value is clinically relevant since the approach permitted surgeons to identify tumors that they would otherwise have been missed (Meijer, EJSO, 2021, DOI: 10.1016/j.ejso.2020.10.034). This point and reference has been added to the revised manuscript under the discussion section.

Line 113:  I understood Authors sampled every hour. What is the max signal at 0.5h? I mention this not to re-do the experiments but to rephrase your finding more carefully. Max signal was at 1h, because the authors did not look earlier?

Line 117-118: Now it seems that the authors did sample more often than every hour. Why is this not plotted? Or how suddenly come values at 15 minutes?

  • We measured fluorescence intensity at 15 minutes and 30 minutes in prior experiments (Lwin, JSO, 2021, DOI: 10.1002/jso.26623 and unpublished data). In the present study, focused on the fluorescence signal hourly, up until 12 hours.  

Figure 1:

  • Why is the background signal “0” at time 0? Background signal should be present always if time 0 means signal before injection of the fluorescence Nb. Strictly speaking fluorescence signal of all three categories, kidney, tumour and background, should be higher than “0” before nanobody injection.
    • To avoid confusion, the time 0 timepoint has been deleted and Figure 1 and 1b have been revised for clarity.
  • It will be necessary that all experiments were performed with the same imaging acquisition settings, e.g., excitation energy and signal collection (exposure and/or gain of detectors)
    • All imaging experiments were performed in a light tight box with the same imaging acquisition settings (ie: excitation, emission channels, and exposure and/or gain of detectors). This point has been added to the revised manuscript in the methods section.
  • Why does the background signal go down in time? Your TBR shows no change in time of the ratio fluorescence to the background. Could it be that this is bleaching, or drying? Would it not be more sensitive to subtract the background signal at each time point? Resulting in only a fluorescence component? This will show that the “real” fluorescent signal remains very constant over time. Image examples from first timepoint, mid and
  • The background signal decreases as the unbound probe is eliminated via renal excretion over time. Both the tumor and the background decrease in signal oover time, but there is a more rapid decrease in the background signal compared to the tumor signal which leads to an initial increase in TBR in the first 6 hours. TBR is preferred over the subtraction of the background fluorescence intensity to provide information on contrast. This information has been added to the revised manuscript under discussion.
    • Image of subcutaneous tumors at each time point have been to the manuscript under the revised Figure 1.
  • An essential missing control is to monitor the same parameters but using a Control-nb-800. I expect the Kidney to display something similar to the aCea-nb-800, while the tumour should be lower and background comparable. Only then do you show a specific increase of signal at the tumour using the aCea-nb-800.
    • We plan to monitor aCtrl nb throughout the entire 12 hours in future experiments.  
  • It is a sound scientific practice to add in figure legends the type of error displayed (in this case is Stdv, but the reader will also like to know the number of independent experiments to better judge the distribution of the variance.
    • Imaging for mice with orthotopic tumors was performed as a single experiment. Imaging for mice with subcutaneous tumors over 12 hours was performed as three separate experiments.
  • Figure 1b how are these error bars calculated? There are specifics rules on how to deal when doing arithmetic with errors. Again, this should be explained/described. Looking at this curve, little can be concluded as errors are so large that there is basically no difference between any time point, thus no conclusions can be made from the graph as presented now.
    • Error bars in previous figures 1a and 1b were calculated using standard deviation. These error bars have been edited using standard error under revised figure 1b and figure 1c.
    • Error calculations are described under the revised methods section of the manuscript: Data is represented as mean ± standard deviation. The maximal fluorescence intensity (MFI) and the TBR were plotted over time, error bars represent standard error of the mean (SEM = standard deviation / Sqrt (counts)).

Line 136-138: Are those values of a single mouse/measurement? If these are the values of the examples shown in Figure 2, could you show the reader what regions were selected for the analysis as “tumour” and “background” signals? A small dotted line showing the ROI will be helpful.

  • The values for the orthotopic xenograft models are calculated from n=4 mice. A supplemental figure indicating the regions selected for analysis as “tumor,” “background,” and “kidney” signals has been added to the revised manuscript.

I understand that it might not be easy to use patient materials to generate an orthotopic xenograph, but a single experiment is still scientifically problematic, this could be chance, luck or artifactual.  If this is the case, I will suggest toning down the title and conclusions, since this has been shown only once, and even here with some questions.

  • As suggested by the reviewer we have modified the conclusion and discussions to reflect this. We have changed the title to: Fluorescent anti-CEA nanobody for rapid tumor targeting and imaging in mouse models of pancreatic cancer.

Figure 3:

  • White light imaging? This looks more like a grey level of the fluorescence image. Please double-check, and if this is the same image in another color scheme is not informative.
    • The “white light” imaging refers to the excitation and emission spectra that the image was captured in. This is explained in the revised legend.
  • How do readers know that the arrow point to the tumour? Because the fluorescence signal is there? This seems to be a tautological explanation. In the images, there are other regions also rather bright, not designated as tumours, what are the criteria? Immunofluorescence or other methods is needed here to demonstrate that the fluorescence obtained by the nanobody coincides with the tumour using another biomarker.
    • The tumor is located where it was surgically implanted. We have performed this approach in similar patient derived orthotopic xenograft systems using anti-CEA antibodies and extensively compared with histology and immunohistochemistry (Lwin, Oncotarget, 2018, DOI: 10.18632/oncotarget.26484). We can perform similar staining in future studies as suggested by the reviewer.
  • Images need scale bars.
    • Reference scale bars have been added to the images.
  • The only difference between “Fluorescence imaging” and “Fluorescence intensity” panels is the lookup table or LUT, but both are images and display fluorescence intensities. The “royal” lookup table only makes it easier to “see” the differences in intensity when compared to the “green” LUT. Good practice will be to display the LUT range beside the image to understand the colour-coding.
    • The reference LUT has been added to the figure.

Line 170-171: Please rephrase this statement. The control nanobody did show a “specific” fluorescence signal at the tumour, it was just a lower signal (Figure 3 even has a dedicated arrow). Unfortunately, this (control nanobody) was not tested in the initial experiments performed for Figure 1, which would have helped here in the discussion.

  • We have revised the sentence to state: “The probe had a higher contrast compared to the control fluorescent nanobody.” This is stated in the discussion section of the revised manuscript.

Line115: Once again, reference to a 15-minute signal that was never shown within the manuscript.

  • We have removed reference to the 15 minute time point from the revised manuscript.

Round 2

Reviewer 2 Report

The authors have partially improved the manuscript.